# Bootstrap Your Own Noise: Denoising Adaptive Noise in Diffusion Models for SSL

## Abstract

We introduce **B**ootstrap **Y**our **O**wn **N**oise (**BYON**), a self-supervised pretraining framework that unifies denoising diffusion with uncertainty-guided contrastive learning to enhance both local and global feature representations. BYON forms a self-reinforcing loop: contrastive learning improves reconstruction quality, and in turn, improved reconstructions refine feature alignment. A Semantic Uncertainty Estimation (SUE) module adaptively reweights contrastive updates based on reconstruction quality, while an Image-specific Adaptive Noise (IAN) adaptively modulates the noise intensity at the image level based on token saliency, perturbing more informative images more strongly. BYON consistently boosts performance on image classification, semantic segmentation, object detection, instance segmentation, and fine-grained visual classification (FGVC) tasks. To ensure reproducibility, the **code** is available in the Supplementary material.

## 1 Introduction

Self-supervised learning (SSL) is a promising paradigm for pre-training large-scale, data-hungry deep networks. By exploiting unlabeled datasets, SSL learns robust, transferable representations that perform strongly on downstream tasks with limited labels. Following the appreciable success of pre-training Transformers with Masked Language Modeling (MLM) (Radford et al., 2018; Devlin et al., 2018; Liu et al., 2019; Clark et al., 2020; Raffel et al., 2020) in natural language processing (NLP), Masked Image Modeling (MIM) (Bao et al., 2022; He et al., 2022; Xie et al., 2022) approaches to mask and predict the portion of an image have become a dominant self-supervised pre-training framework in computer vision. The simplicity and effectiveness of MIM have made it a prominent choice for self-supervised pre-training, showing impressive results in downstream tasks such as image classification, semantic segmentation, and object detection.

Building upon the success of MIM, recent work explores integrating diffusion models into self-supervised learning, forming a new paradigm for representation learning. Denoising-based pre-training (Wei et al., 2023; Zheng et al., 2023) augments MIM with generative denoising to capture finer local structure beyond masked patch reconstruction. By introducing a progressive denoising process (Rombach et al., 2022; Ramesh et al., 2021; Saharia et al., 2022), these approaches aim to enrich feature learning and potentially improve transferability across a wide range of recognition tasks, including image classification, semantic segmentation, object detection, instance segmentation, and fine-grained visual classification.

While effective for local feature learning, MIM and diffusion-based pre-training can underutilize global semantic structure in practice. MIM (Bao et al., 2022; He et al., 2022; Xie et al., 2022) reconstructs masked regions largely from nearby context, which encourages locality but lacking global coherence; diffusion-based approaches (Wei et al., 2023; Zheng et al., 2023) progressively refine high-level visual representations through denoising but lack explicit alignment of global feature distributions across image instances. As suggested by Fig. 1, attention-distance profiles and head-diversity measures skew toward shorter ranges with depth, indicating weaker long-range aggregation relative to local cues. Thus, our analyses indicate a **tendency toward local bias in the absence of explicit global alignment objectives**.

To mitigate these tendencies, we explore integrating contrastive learning (Oord et al., 2018; Bachman et al., 2019; Chen et al., 2020b; He et al., 2020; Grill et al., 2020; Chen & He, 2021) with

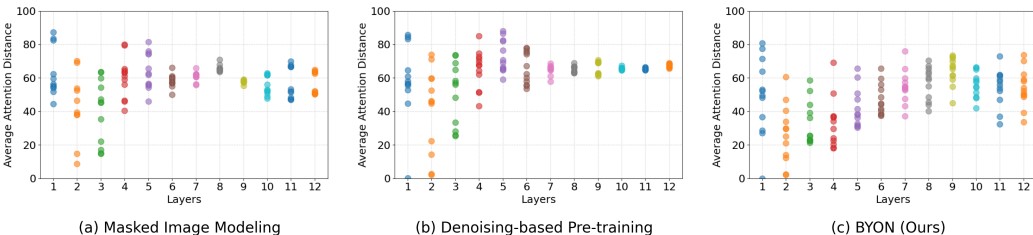

(a) Masked Image Modeling          (b) Denoising-based Pre-training          (c) BYON (Ours)

Figure 1: **Average attention distance across heads** (dots) with respect to layer depth for (a) Masked Image Modeling (Xie et al., 2022), (b) Denoising-based Pre-training (Wei et al., 2023), and (c) our proposed method (BYON), all with ViT-B. In (a) and (b), early layers cover a range of distances but remain biased toward local attention (see the darker average dots). As layer depth increases, they fail to aggregate information across a broader spatial distribution. BYON shows more varied per-head behaviors and a more balanced distance distribution across layers, suggesting a better mix of local and global representation learning.

denoising-based pre-training to strengthen both local and global representations. Contrastive learning, the dominant paradigm in SSL before the rise of MIM, has been effective at organizing the global feature space via instance-level discrimination, and, through bootstrapping (Grill et al., 2020), encourages consistency across views of the same image. These observations suggest that a contrastive objective alongside denoising could provide an explicit global alignment signal that complements local reconstruction.

Building on this insight, we propose **B**ootstrap **Y**our **O**wn **N**oise (**BYON**), a self-supervised pre-training framework that couples contrastive learning with denoising within the MIM setup, bootstrapping representations from noised inputs. BYON pairs the fine-grained local detail encouraged by diffusion-style reconstruction with an explicit contrastive objective that aligns instance-level embeddings, promoting more coherent global structure. Such global alignment can improve semantic transfer by stabilizing cross-view invariances and reducing spurious locality. In practice, this local–global coupling yields representations better suited to diverse recognition tasks, where preserving semantic structure is typically paramount.

We further leverage reconstruction uncertainty as a guiding signal to integrate diffusion-based reconstruction with contrastive learning. Specifically, a Semantic Uncertainty Estimation (SUE) module quantifies per-image reconstruction accuracy and dynamically reweights the contrastive loss: higher uncertainty down-weights its contribution to the contrastive learning as it indicates a greater semantic gap between the reconstructed and original features, while lower uncertainty increases it, reinforcing learning for well-aligned representations. By leveraging this interaction, our method creates a self-reinforcing feedback loop where contrastive learning benefits from reliable reconstruction, and in return, contrastive learning enhances the semantic alignment of diffusion-recovered features, improving overall representation learning capability.

Lastly, we introduce Image-specific Adaptive Noise (IAN) to enhance the diffusion-based pretraining process. Unlike prior methods that apply uniform random noise (Wei et al., 2023; Zheng et al., 2023), IAN adjusts image-level noise intensity based on token saliency: saliency scores are computed per token, and assign stronger noise perturbations to images with more salient tokens. This biases the model toward informative content by encouraging reconstruction of essential features during denoising.

Our local–global framework (BYON) yields consistent gains across standard benchmarks (Deng et al., 2009b; Zhou et al., 2017a; Lin et al., 2014a), covering image classification, semantic segmentation, object detection, and instance segmentation. Relative to diffusion-based pretraining (Wei et al., 2023; Zheng et al., 2023), BYON improves ImageNet-1K top-1 by accuracy by 0.9%, semantic segmentation by 1.8%, and detection/instance segmentation by up to 7.5%. BYON also performs strongly on fine-grained recognition benchmarks (CUB-200-2011 (Wah et al., 2011), NABirds (Van Horn et al., 2015), iNaturalist 2017 (Van Horn et al., 2017), iNaturalist 2018 (Van Horn et al., 2018), Stanford Cars (Krause et al., 2013), and Aircraft (Maji et al., 2013)), with gains up to 4.3%. These results indicate that coupling diffusion-style local cues with explicit global alignment produces more transferable representations.

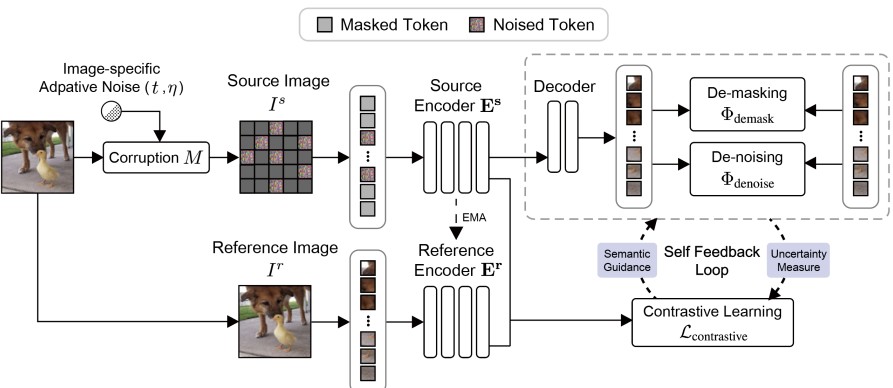

Figure 2: **Overview of Bootstrap Your Own Noise (BYON)**. The framework integrates denoising diffusion models with uncertainty-guided contrastive learning to enhance both local and global feature representations. BYON consists of three key components: (1) Image-specific Adaptive Noise (IAN), (2) the Semantic Uncertainty Estimation (SUE) module, and (3) bootstrapping representations from the diffusion model. This process forms a self-reinforcing feedback loop, where better-aligned features improve reconstruction, which in turn refines uncertainty estimation and strengthens contrastive learning.

We summarize our contributions:

- We introduce self-supervised pre-training framework (BYON) that couples diffusion-style reconstruction with contrastive learning, unifying local and global representation learning for transfer.
- The Semantic Uncertainty Estimation (SUE) module measures reconstruction reliability as a guidance signal in self-reinforcing feedback loop.
- We propose Image-specific Adaptive Noise (IAN), which adjusts noise levels per image based on token saliency scores, prioritizing the reconstruction of critical visual information.

## 2 RELATED WORK

### 2.1 CONTRASTIVE LEARNING

Contrastive learning—spanning pairwise and similarity-based objectives (He et al., 2020; Chen et al., 2020b; Grill et al., 2020)—learns instance-discriminative representations by pulling positives and pushing negatives. Extensions reduce reliance on explicit negatives via momentum encoders or stop-gradient (Grill et al., 2020; Chen et al., 2020b). Inspired by this line, BYON bootstraps diffusion outputs to couple fine-grained local learning (denoising) with uncertainty-guided global alignment.

### 2.2 MASKED IMAGE MODELING (MIM)

Inspired by the scalability of MLM in NLP (Devlin et al., 2018; Radford et al., 2018; 2019; Liu et al., 2019; Brown et al., 2020), MIM learns by predicting missing image content. Early work (Context Encoder) regressed missing pixels with CNNs (Pathak et al., 2016). With Transformers, attention-based MIM achieved strong results (Chen et al., 2020a; Dosovitskiy et al., 2020; Bao et al., 2022; Zhou et al., 2022; He et al., 2022; Xie et al., 2022; Dong et al., 2022); e.g., BEiT uses a DALL·E-style tokenizer (Bao et al., 2022), iBOT jointly updates it via momentum (Zhou et al., 2022), while MAE and SimMIM simplify training with lightweight decoders (He et al., 2022; Xie et al., 2022). Several works combine MIM with contrastive signals (Zhou et al., 2022; Kakogeorgiou et al., 2022; Li et al., 2021), and recent studies highlight masked tokens' role in convergence and accuracy (Choi et al., 2024b; 2025). Our method follows the MIM paradigm but extends it with diffusion-driven denoising and uncertainty-guided local–global discrimination, aiming to refine fine-grained features and align global semantics.

## 2.3 DENOISING DIFFUSION MODELS (DDMs)

DDMs corrupt and restore signals via iterative noise injection and denoising, yielding rich latent features useful for fine-grained recognition. Recent work integrates diffusion with SSL by mixing masking and noising (Wei et al., 2023; Zheng et al., 2023), but adding denoising to MIM alone has offered limited gains. We propose to bootstrap representations from the denoising process, using progressive refinement for local detail while coupling it with global semantic alignment, to improve transferability.

## 2.4 UNCERTAINTY ESTIMATION

MIM-style regression yields pixel estimates, not distributions, motivating uncertainty signals for reliability. While variance predictors exist (Lakshminarayanan et al., 2017; Kendall & Gal, 2017), they add compute. Instead, we exploit the intact image already available in reconstruction: we define a distance-based proxy for uncertainty between corrupted-patch encodings and the intact target, and use it in the Semantic Uncertainty Estimation (SUE) module to guide local–global discrimination.

## 3 METHOD

### 3.1 OVERALL FRAMEWORK

In this section, we introduce Bootstrap Your Own Noise (BYON), a novel self-supervised learning framework that integrates denoising diffusion models and uncertainty-guided contrastive learning to enhance both local and global feature representations. Our method consists of three key components: Semantic Uncertainty Estimation (SUE), Image-specific Adaptive Noise (IAN), and bootstrapping representations where the denoising diffusion and contrastive learning are conducted in a complementary manner.

The overall architecture is illustrated in Fig. 2. A source image $I^s$, consisting of noisy visible tokens and masked tokens, is generated by applying masking and noise to an original image $I$. Here, no corruption is applied to a reference image $I^r$, *i.e.*, $I^r = I$. The source and target images are encoded into full feature maps as well as class tokens. Formally, given encoders $\mathbf{E}^s$ and $\mathbf{E}^r$, $F^s = \mathbf{E}^s(I^s)$, $F^r = \mathbf{E}^r(I^r)$, where $F^s, F^r$ denote the token-wise feature maps. We also extract the associated class tokens (global summaries) $z_0^s, z_0^r \in \mathbb{R}^D$: $z_0^s = \mathrm{CLS}(F^s), z_0^r = \mathrm{CLS}(F^r)$. Here, the reference encoder $\mathbf{E^r}$ is updated via an exponential moving average (EMA) of the source encoder $\mathbf{E^s}$ to maintain stable representation learning. The decoder then takes as inputs the source feature maps from the source encoder and reconstructs an original image.

BYON bootstraps representations from the denoising process of diffusion models, enforcing global feature alignment through contrastive learning while its contribution is dynamically reweighted based on uncertainty of image reconstruction (SUE). This process establishes a self-reinforcing feedback loop, where improved global alignment enhances image reconstruction, which in turn reinforces contrastive learning for global feature alignment. Furthermore, IAN dynamically adjusts noise levels based on the saliency of an input image $I$ to enhance the diffusion-based pretraining process.

### 3.2 SEMANTIC UNCERTAINTY ESTIMATION (SUE)

We first introduce a new method, Semantic Uncertainty Estimation (SUE), which measures the uncertainty of image reconstruction. Since early epochs exhibit inaccurate reconstruction, bootstrapping latent representations becomes unreliable, making it crucial to assign appropriate importance to contrastive loss based on the image-level reliability. To this end, the SUE predicts the uncertainty of the reconstructed image and reweights the contrastive loss.

We define the image-level uncertainty score $U$ based on the divergence between the reconstructed image $\hat{I}$ and the original image $I$.

$$U = \sigma(d(\hat{I}, I)), \qquad \sigma(k) = \frac{1 - e^k}{1 + e^k}, \qquad (1)$$

where $d$ represents the L1 distance between two inputs. The obtained uncertainty map $U$ is then truncated with a threshold $\tau$ as

$$\delta^u = \sum_i \mu_i/N, \qquad \mu_i = (U_i < \tau). \tag{2}$$

$N$ is the number of tokens. We used a fixed value $\tau = 0.5$ for all experiments. Accordingly, the hard thresholded map $\mu$ indicates reliable tokens, which will be used in the following section. The estimated $\mu$ is used to reweight the contrastive loss, ensuring that more reliable reconstructions contribute more to global feature alignment.

### 3.3 Bootstrapping Representations

#### 3.3.1 Global Contrastive Learning using SUE

To enforce global feature alignment, we apply contrastive learning between the classification tokens from the source and reference views. As illustrated in Fig. 3, the contrastive module consists of a source projection head, a source prediction head, and a reference projection head, with only the source encoder and projection head being updated during train-

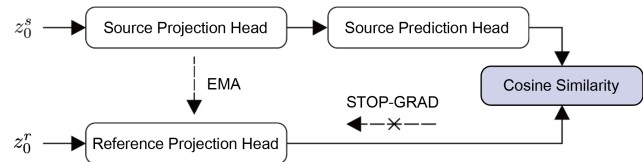

Figure 3: **Global Contrastive Learning.** The local and global discrimination learning is performed using the asymmetric heads $\phi^s$ and $\phi^r$.

ing. Early training stages produce noisy reconstructions, making direct bootstrapping unreliable. To mitigate this, we estimate uncertainty of image reconstruction and use it to dynamically weight the contrastive loss. The contrastive loss is reweighted based on the computed uncertainty map $\mu$ in equation 2 as follows:

$$\mathcal{L}_{\text{contrastive}} = -\delta^u \frac{< \phi^s(z_0^s), \phi^r(z_0^r) >}{|\phi^s(z_0^s)|_2 |\phi^r(z_0^r)|_2}, \tag{3}$$

where $\phi^s$ is the source projection and prediction heads, while $\phi^r$ represent the reference projection head applied to the classification tokens.

#### 3.3.2 Self-Reinforcing Feedback Loop

As illustrated in Fig. 2, the proposed bootstrapping framework forms a self-reinforcing feedback loop. Specifically, contrastive learning encourages global feature alignment, which in turn enhances reconstruction quality. The improved reconstruction facilitates more precise uncertainty estimation, and the refined uncertainty further strengthens contrastive learning. This iterative synergy between uncertainty-aware representation bootstrapping and contrastive learning enables BYON to learn stable, transferable global representations, leading to improved performance in downstream tasks.

### 3.4 Image-specific Adaptive Noise (IAN)

Existing diffusion-based pre-training methods (Wei et al., 2023; Zheng et al., 2023) apply uniform random noise to images, which often fails to emphasize critical visual information. To address this limitation, we introduce Image-specific Adaptive Noise (IAN), which dynamically adjusts the noise level based on the importance of individual input image, which is computed using token saliency scores. This mechanism assigns stronger perturbations where it matters most, encouraging the model to prioritize essential features during denoising.

#### 3.4.1 Token Saliency Score

To determine the importance of each token, we compute token saliency scores using the outgoing attention weights (Choi et al., 2025) from the self-attention mechanism used in the Transformer. Given a source input token sequence $X \in \mathbb{R}^{N \times D}$, the affinity matrix $A \in \mathbb{R}^{N \times N}$ is computed as $A = XX^T$. Applying the softmax function along each row normalizes the attention scores:

$$\hat{A}_{i,j} = \frac{e^{A_{i,j}}}{\sum_k e^{A_{i,k}}}. \tag{4}$$

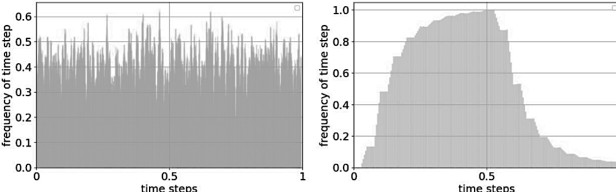

Figure 4: **Comparison of noise distribution**. Prior diffusion-based pre-training (Wei et al., 2023; Zheng et al., 2023) applies uniform random noise, yielding unstructured patterns. IAN scales noise by image-level saliency (from token scores), assigning stronger perturbations to informative images and biasing reconstruction toward meaningful regions.

The token saliency score $S \in \mathbb{R}^N$ is then obtained by summing the outgoing attention weights across all tokens:

$$S_j = \sum_i \hat{A}_{i,j} \tag{5}$$

where higher values indicate a greater influence on the overall image representation. To introduce diversity, we apply randomized perturbation to the saliency scores, as $\tilde{S}_j = S_j + \mathcal{U}(0, 0.5)$ where $\mathcal{U}(0, 0.5)$ is sampled from a uniform distribution.

### 3.4.2 IAN BASED ON TOKEN SALIENCY SCORES

We utilize the saliency scores $\tilde{S}$ to determine the noise level applied to each token. The adaptive noise level is computed as:

$$y = \frac{1}{N} \sum_j \mathbb{1}(\tilde{S}_j > \delta). \tag{6}$$

$$t = \min(T_{max}, \max(T_{min}, \lambda \cdot y + d_{\text{noise}})), \tag{7}$$

$$\eta = \Delta\eta \times \frac{1}{N} \sum_j \tilde{S}_j + \frac{1}{N} \sum_j \mathcal{N}(0, 1). \tag{8}$$

where saliency-based thresholding $y$ determines how many tokens exceed a given threshold $\delta$. We set $T_{min}$ and $T_{max}$ as the minimum and maximum timesteps and draw a stochastic term $d_{\text{noise}} \sim \text{Uniform}(0, 1)$. A scaling coefficient $\lambda$ controls how strongly $y$ influences the assigned timestep $t$, enabling adaptive scheduling. Adaptive noise level $\eta$ is scaled according to the normalized saliency score and further perturbed by a small random Gaussian noise term, where $\Delta\eta$ is a hyperparameter that controls the base noise intensity. This ensures that tokens with higher saliency receive stronger noise perturbations, forcing the model to focus on reconstructing essential visual features, as depicted in Fig. 4.

As BYON is built upon the MIM framework, it follows the standard masked image modeling approach while applying ISN. Thus, BYON applies both token masking and adaptive noise to the source image before feeding it into the source encoder. Given a source input image $I^s \in \mathbb{R}^{N \times D}$, we define a binary mask matrix $M \in \{0, 1\}^N$, where $M_i = 1$ indicates a noisy visible token and $M_i = 0$ a masked token for $i = 1, ..., N$. The final corrupted source token $\tilde{x}$ is then generated by applying masking and injecting IAN into the input token $x$ as follows:

$$\tilde{x} = \left(\sqrt{\alpha_t} \cdot x + \sqrt{1 - \alpha_t} \cdot \eta \odot \epsilon\right) \odot M + \theta \odot (1 - M), \tag{9}$$

$$\epsilon \sim \mathcal{N}(0, 1). \tag{10}$$

where $\mathcal{N}(0, 1)$ is a standard Gaussian noise function, $\theta$ is a learnable masked token embedding, and $\eta$ is the saliency-based adaptive noise level. This adaptive noise strategy forces the model to learn more robust local representations.

### 3.5 RECONSTRUCTION BY DE-NOISING AND DE-MASKING

The source feature is passed through a lightweight decoder that performs two key reconstruction processes: de-noising and de-masking. These processes aim to recover the original feature representations. As shown in Fig. 2, the decoder receives noisy and masked representations and reconstructs them through the following two functions:

$$\hat{x}_n^{t-1} = \Phi_{\text{denoise}}(x_n^t, x_m, t) \quad \hat{x} = \Phi_{\text{demask}}(x_m, x_v), \tag{11}$$

Figure 5: **Self-attention from BYON.** We visualized the self-attention of the image classification token on the last layer. BYON explicitly encodes the local fine-grained semantics altogether with the global semantics, resulting in favorable performance improvement. For instance, BYON captures semantics from large instances like reptiles to very fine semantics like spider legs.

where $x_n^t$ represents noisy visible tokens at timestep $t$, $x_m$ represents masked tokens, and $x_v$ represents visible (unmasked) tokens.

The denoising process is modeled as a function $\Phi_{\text{denoise}}$, which predicts the token representations $\hat{x}_m^{t-1}$ from their noisy counterparts. This process follows the diffusion model's iterative refinement, progressively removing noise while leveraging context from the unmasked tokens. This ensures that high-frequency details are preserved while gradually refining local structures.

Parallel to the de-noising step, the de-masking process reconstructs masked tokens by utilizing surrounding visible token embeddings. The function $\Phi_{\text{demask}}$ predicts the reconstructed representation $\hat{x}$. This process enables the model to restore the missing semantic information by leveraging neighboring token structures. Refer to Appendix L for detailed definitions of objectives.

**Total Loss.** The overall learning objective is formulated as:

$$\mathcal{L}_{\text{total}} = \mathcal{L}_{\text{demask}} + \lambda \cdot \mathcal{L}_{\text{denoise}} + \mathcal{L}_{\text{contrastive}} \qquad (12)$$

where $\lambda$ is a loss-balancing hyperparameter that controls the relative importance of denoising loss in the overall objective. In all experiments, we set $\lambda = 0.1$ to maintain consistency.

## 4 EXPERIMENTS

### 4.1 COMPARISON METHODS

To ensure a fair comparison, we compare BYON against state-of-the-art denoising-based pre-training methods (Wei et al., 2023; Zheng et al., 2023) explicitly designed for recognition tasks. Other recent denoising-based methods (Peebles & Xie, 2023; Gao et al., 2023; Hatamizadeh et al., 2024) are not designed for recognition tasks and demonstrated substantially lower performance in preliminary evaluations, making their inclusion neither informative nor relevant to our recognition-focused pre-training objectives. Furthermore, as BYON builds upon MIM, we additionally benchmark against two canonical MIM methods, SimMIM (Xie et al., 2022) and MAE (He et al., 2022), to assess improvements beyond standard MIM architectures. To ensure the validity of our findings, all methods are **reproduced under identical hardware and training configurations** to facilitate a controlled and unbiased comparison. The baselines have been trained in large cluster resources that are not available to everyone. Please note that all comparisons used the same setup, with code available for verification. For methods with publicly available code (Xie et al., 2022; He et al., 2022; Zheng et al., 2023), we use the official implementations, while for those without (Wei et al., 2023), we reimplement them based on the original papers. The difference in reproduced performance stems from hardware differences.

### 4.2 IMPLEMENTATION DETAILS

All experiments were conducted under identical conditions for a fair analysis, with each method reimplemented, leading to potential deviations from reported results in original papers. We used ViT-B (Dosovitskiy et al., 2020) as the backbone and trained all models for 400 epochs on ImageNet-1K (Deng et al., 2009b) using $4 \times$ A100 GPUs. Following comparison methods (Xie et al., 2022; He et al., 2022; Wei et al., 2023; Zheng et al., 2023), ImageNet-1K image classification dataset (Deng et al., 2009a) was used without label information in the pre-training step. Full implementation details, including code and experimental settings, are provided in the Supplementary material.

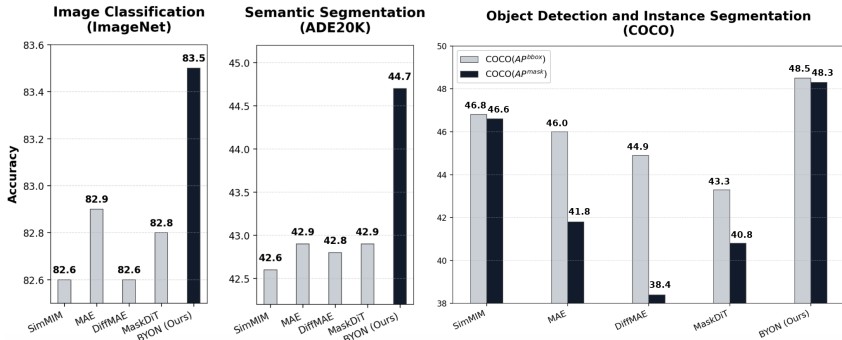

Figure 6: **Evaluation on Image Classification, Semantic Segmentation, Object Detection, and Instance Segmentation.** BYON surpasses all baselines across tasks: it attains 83.5% top-1 accuracy on ImageNet-1K (Deng et al., 2009b), 44.7% mIoU on ADE20K (Zhou et al., 2017a), and 48.5 $AP^{bbox}$ / 48.3% $AP^{mask}$ on COCO (Lin et al., 2014a).

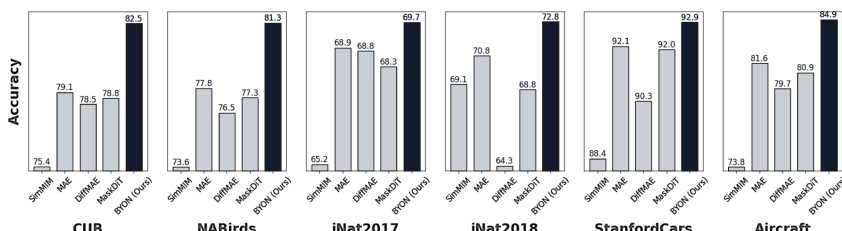

Figure 7: **Evaluation on Fine-Grained Visual Classification (FGVC).** BYON consistently outperforms all baselines in FGVC tasks, demonstrating its ability to capture both fine-grained local details and global semantic consistency.

### 4.3 VISUALIZATION OF ATTENTION MAPS

To better understand how BYON enhances feature learning, we visualize the attention maps of the model trained with our framework. Fig. 5 presents the input images (top row) and their corresponding attention responses (bottom row). Our method effectively encodes both local fine-grained semantics and global contextual information, leading to improved representation learning. In alignment with Fig. 1, BYON attends to critical regions across various object scales from large instances like reptiles to fine details such as spider legs, demonstrating its ability to capture rich semantic structures for enhanced feature alignment and recognition performance.

### 4.4 EVALUATION ON IMAGE CLASSIFICATION

We fine-tuned our pre-trained model on ImageNet-1K dataset (Deng et al., 2009a) using an AdamW optimizer for 100 epochs. In Fig. 6, BYON achieves 83.5% top-1 accuracy, outperforming all baselines. Compared to MaskDiT (82.8%) and DiffMAE (82.6%), which also leverage diffusion-based pre-training, BYON demonstrates superior feature learning by effectively integrating local-global representation learning through uncertainty-guided contrastive learning.

### 4.5 EVALUATION ON SEMANTIC SEGMENTATION

We fine-tuned the pre-trained model on ADE20K dataset (Zhou et al., 2017b) which consists of 25K images of 150 semantic categories. The semantic segmentation performance was measured with mean intersection over union (mIOU) in Fig. 6. BYON attains a 44.7% mIoU, significantly surpassing all baselines, with a notable 1.8% absolute improvement over MaskDiT and MAE (42.9%). This highlights BYON's ability to transfer more structured representations to dense prediction tasks, where global contextual understanding is crucial.

| De-noising (Noised Tokens) | De-masking (Masked Tokens) | Contrastive learning | IAN (w/ Noised Token only) | SUE (w/ Contrastive Learning only) | Acc |
|:---:|:---:|:---:|:---:|:---:|:---:|
| | | ✓ | | | 80.27 |
| ✓ | | | | | 80.14 |
| ✓ | | | ✓ | | 82.38 |
| ✓ | | ✓ | | ✓ | 82.01 |
| | ✓ | | | | 82.89 |
| | ✓ | ✓ | | ✓ | 83.02 |
| ✓ | ✓ | | | | 82.86 |
| ✓ | ✓ | | ✓ | | 83.16 |
| ✓ | ✓ | ✓ | | ✓ | 82.84 |
| ✓ | ✓ | ✓ | ✓ | ✓ | 83.56 |

Table 1: **Ablation over all component combinations.** IAN consistently helps when noised tokens are used; SUE contributes most with IAN and both tasks; the full model performs best, supporting local–global coupling with adaptive corruption.

### 4.6 EVALUATION ON OBJECT DETECTION AND INSTANCE SEGMENTATION

To transfer the pre-trained model to object detection and instance segmentation, we fine-tuned on the COCO dataset (Lin et al., 2014b) using Mask R-CNN (He et al., 2017). Fig. 6 presents the results in terms of bounding box AP ($AP^{bbox}$) and mask AP ($AP^{mask}$). BYON achieves 48.5% $AP^{bbox}$ and 48.3% $AP^{mask}$, outperforming all baselines. Compared to SimMIM (46.8% / 46.6%) and MAE (46.0% / 41.8%), BYON demonstrates improved transferability, particularly in dense prediction tasks, where both local and global information are crucial. Moreover, BYON surpasses DiffMAE (44.9% / 38.4%) and MaskDiT (43.3% / 40.8%), highlighting the benefits of uncertainty-guided contrastive learning and adaptive noise in enhancing object-centric representations.

### 4.7 EVALUATION ON FINE-GRAINED VISUAL CLASSIFICATION (FGVC)

To assess the effectiveness of BYON in Fine-Grained Visual Classification (FGVC) tasks, we evaluate its performance on diverse FGVC benchmarks, including CUB-200-2011 (Wah et al., 2011), NABirds (Van Horn et al., 2015), iNaturalist 2017 (Van Horn et al., 2017), iNaturalist 2018 (Van Horn et al., 2018), Stanford Cars (Krause et al., 2013), and Aircraft (Maji et al., 2013).

As shown in Fig. 7, BYON demonstrates consistent superiority across all FGVC benchmarks, suggesting that its local-global feature learning strategy plays a crucial role in distinguishing fine-grained patterns. Notably, DiffMAE (Wei et al., 2023) and MaskDiT (Zheng et al., 2023), which rely solely on denoising-based pretraining, fail to bridge the gap between generative learning and discriminative tasks, leading to suboptimal performance. In contrast, BYON effectively leverages contrastive learning to enforce global feature alignment, allowing it to better separate visually similar categories while still benefiting from the fine-grained representation learning of diffusion models. These results highlight that BYON's balanced approach to local and global representation learning is particularly well-suited for fine-grained recognition, offering a compelling alternative to existing pretraining paradigms.

### 4.8 ABLATION STUDY

We evaluate all component combinations—De-noising (noised tokens), De-masking (masked tokens), IAN (saliency-adaptive noise; applicable only with De-noising), and SUE (uncertainty-guided reweighting). IAN consistently helps whenever noised tokens are present; SUE is most effective when paired with IAN and both tasks; the full configuration yields the highest accuracy, supporting a design that couples local (denoising) and global (alignment) signals with adaptive corruption.

## 5 CONCLUSION

We introduced BYON, a self-supervised framework that integrates diffusion models with uncertainty-guided contrastive learning. BYON refines feature alignment through a self-reinforcing loop, leveraging SUE for adaptive contrastive weighting and IAN for saliency-driven noise. Experiments confirm its effectiveness across diverse tasks.

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

# Appendix

## A    MULTI-SEED EVALUATION

|  | **SimMIM** | **MAE** | **DiffMAE** | **MaskDiT** | **BYON** |
|---|---|---|---|---|---|
| ImageNet | $82.54 \pm 0.15$ | $82.78 \pm 0.20$ | $82.60 \pm 0.33$ | $82.86 \pm 0.05$ | $83.56 \pm 0.16$ |
| ADE20K | $42.62 \pm 0.16$ | $43.02 \pm 0.12$ | $42.68 \pm 0.48$ | $42.90 \pm 0.11$ | $44.46 \pm 0.21$ |

Table 2: Comparison on ImageNet top-1 (%) and ADE20K mIoU (%). BYON achieves the best mean $\pm$ std in both settings.

We evaluate all methods over five independent runs with different random seeds and report mean and standard deviation in Tab. 2, thereby assessing both central tendency and run-to-run stability. On ImageNet, BYON attains $83.56 \pm 0.16$, exceeding SimMIM (Xie et al., 2022), MAE (He et al., 2022), DiffMAE (Wei et al., 2023), and MaskDiT (Zheng et al., 2023). The standard deviation remains low and comparable to MIM baselines, while DiffMAE exhibits larger variability ($\pm 0.33$), indicating reduced stability. On ADE20K, BYON reaches $44.46 \pm 0.21$, improving over comparison methods. The gains are more pronounced in segmentation. Taken together, the results indicate that BYON delivers higher average performance and favorable stability across runs, clarifying its empirical benefit under a standardized evaluation protocol.

## B    IMPACT OF LONGER PRE-TRAINING DURATION

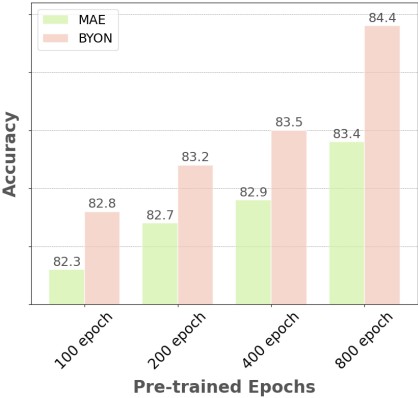

Figure 8: **Impact of Longer Pre-Training Duration.** To further examine the effect of extended training, we conduct a long 800-epoch pre-training experiment, observing additional improvements. While BYON benefits from longer training, our results confirm that 400 epochs provide a well-balanced trade-off between efficiency and performance.

We analyze the effect of pre-training duration by comparing BYON with MAE over different training epochs, as shown in Fig. 8. We demonstrate that BYON achieves performance comparable to MAE (He et al., 2022) trained for 800 epochs with just 400 epochs, validating its efficiency in self-supervised pre-training. This indicates that BYON trained with 400 epochs is sufficient for robust feature learning across benchmarks. To further examine the effect of extended training, we conduct a long 800-epoch pre-training experiment, observing additional improvements. While BYON benefits from longer training, our results confirm that 400 epochs provide a well-balanced trade-off between efficiency and performance.

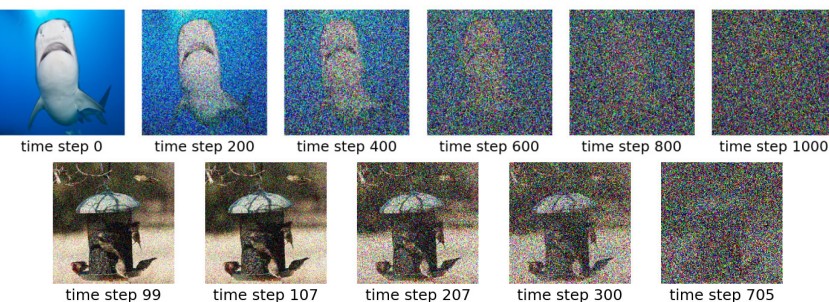

Figure 9: **Comparison of whole timestep and top-k timestep visualizations.** We present a visualization of different timesteps in the denoising process. The top row illustrates images at various timesteps ranging from 0 to 1000, showing the progressive corruption of the original image as noise increases. The bottom row visualizes the top five most frequently selected timesteps in BYON's dynamic noise adaptation mechanism.

## C ANALYSIS OF DYNAMIC NOISE ADAPTATION

Fig. 9 presents a visualization of different timesteps in the denoising process. The top row illustrates images at various timesteps ranging from 0 to 1000, showing the progressive corruption of the original image as noise increases. The bottom row visualizes the top five most frequently selected timesteps in BYON's dynamic noise adaptation mechanism. These timesteps are determined by selecting the highest-ranked values in predefined intervals (e.g., 0–100, 100–200, 200–300), reflecting the model's dynamic preference for specific noise levels during training.

Unlike DiffMAE, which relies on a standard diffusion-based reconstruction process, BYON incorporates a hybrid masking strategy (Zheng et al., 2023; Choi et al., 2024a), introducing an additional challenge when reconstructing masked patches at extreme noise levels. To address this, we modify the diffusion process by reducing the betas values, slowing down the noise diffusion rate. This allows the model to better handle high-noise conditions by ensuring that essential semantic structures are not excessively degraded. By adopting this noise scheduling, BYON enhances its reconstruction capability, particularly in handling partially corrupted patches, ultimately improving its feature learning and transferability across recognition tasks.

## D COMPARISON WITH CONTRASTIVE–MIM METHODS

|  | iBOT | CMAE | BYON |
|---|---|---|---|
| ImageNet | $82.03 \pm 0.17$ | $82.89 \pm 0.20$ | $83.56 \pm 0.16$ |

Table 3: **Comparison with contrastive-MIM.** ImageNet top-1 (%) reported as mean $\pm$ std over 5 seeds.

Table 3 compares BYON to representative contrastive–MIM approaches (Zhou et al., 2022; Huang et al., 2022). Under the same 5-seed protocol, BYON attains $83.56 \pm 0.16$, surpassing iBOT (Zhou et al., 2022) ($82.03 \pm 0.17$; +1.53) and CMAE (Huang et al., 2022) ($82.89 \pm 0.20$; +0.67).

The results indicate that adding a contrastive signal benefits not only MIM-style pre-training (e.g., iBOT, CMAE) but also our diffusion-enhanced setting. In BYON, contrastive learning supplies an explicit global alignment objective that complements the local, reconstruction-driven inductive bias of diffusion. This coupling reduces the tendency toward locality (observed in reconstruction-only regimes), stabilizes cross-view invariances at the instance level, and yields higher ImageNet top-1 with comparable or lower variance. In other words, contrastive learning acts as a structural prior on the feature space that is orthogonal to the denoising objective: the former aligns embeddings across augmentations and instances, while the latter refines fine-grained details via progressive noise removal.

# E    LAMBDA TUNING FOR RECOGNITION

|      | $\lambda=0$ | $\lambda=0.1$ | $\lambda=0.5$ | $\lambda=1$ |
|------|------|------|------|------|
| Acc  | 82.94 | **83.56** | 83.28 | 81.70 |

Table 4: **Lambda tuning for recognition.** Recognition accuracy (%) under different $\lambda$ values. $\lambda=0.1$ provides the best performance.

We sweep $\lambda \in \{0, 0.1, 0.5, 1\}$ to quantify how strongly the saliency signal should shape the noise/timestep schedule. Performance peaks at $\lambda=0.1$ (Tab. 4). Two trends emerge: (i) moving from $\lambda=0$ to $0.1$ adds a mild saliency bias that improves alignment between corruption and informative content, yielding a clear gain; (ii) larger values $(0.5, 1)$ degrade accuracy—over-emphasizing saliency leads to overly aggressive perturbations on salient regions and reduces the diversity of training signals, which harms recognition transfer.

For fairness, we note that MaskDiT (Zheng et al., 2023) originally uses $\lambda=10$ for generation. Because recognition favors conservative corruption (for stable feature transfer), we re-tuned both MaskDiT and BYON to $\lambda=0.1$ in all recognition comparisons. This setting consistently provided the best accuracy in our protocol.

# F    KEY COMPONENTS AND COUPLINGS: RELIABILITY-AWARE ALIGNMENT (CL+SUE) AND SALIENCY-AWARE CORRUPTION (DDM+IAN)

Our framework couples denoising diffusion with contrastive alignment to learn features that are both locally precise and globally coherent. The ablations (Tab. 1) expose two necessary couplings and clarify why decoupled variants underperform.

(1) Contrastive Learning (CL) must be reweighted by Semantic Uncertainty Estimation (SUE).

Reconstruction reliability varies across images and training steps. SUE provides a per-sample reliability score and reweights the CL objective accordingly. Without SUE, CL treats unreliable reconstructions as clean positives, injecting label noise into the alignment target and flattening the instance structure. With SUE, alignment pressure is strong when reconstructions are faithful and weak when they are uncertain, yielding a reliability-aware global signal. In Tab. 1, adding SUE to the denoising path improves accuracy over denoising alone; its effect is largest when paired with IAN and de-masking, indicating that SUE complements, rather than replaces, local reconstruction.

(2) Denoising Diffusion Models (DDM) must be paired with Image-specific Adaptive Noise (IAN).

Denoising is only as instructive as the corruption schedule. Uniform noise produces many easy or miscalibrated training cases; IAN scales noise by saliency so that corruption preferentially targets informative content. This increases the fraction of semantically meaningful reconstruction signals while preserving diversity via stochasticity. In the ablation, DDM+IAN consistently outperforms DDM alone and boosts joint de-noising+de-masking, demonstrating that adaptive corruption is a first-order factor in representation quality.

Together, these two couplings define a training objective that fundamentally differs from prior diffusion-based SSL methods, which rely solely on reconstruction-driven denoising, focusing on local features. In contrast, BYON integrates local reconstruction with global alignment through a coupled objective (SUE+CL and IAN+DDM).

**Design implication.**    The effective unit is not CL or DDM in isolation, but the couples (CL+SUE) and (DDM+IAN). These couples are orthogonal to the underlying MIM backbone: they can be attached to MAE (He et al., 2022)/SimMIM (Xie et al., 2022)/iBOT (Zhou et al., 2022)-style pipelines with minimal changes, as evidenced by consistent gains in Tab. 1. In summary, reliability-aware alignment (CL+SUE) and saliency-aware corruption (DDM+IAN) are the decisive ingredients; together they convert reconstruction signals into transferable representations by balancing local refinement with stable global alignment.

# G    REPORT OF FLOPs AND GPU HOURS

|  | DiffMAE | MaskDiT | BYON |
|---|---|---|---|
| GFLOPs | 45.5 | 43.7 | 63.8 |
| GPU Hours | 84 | 83 | 97 |

Table 5: **Compute comparison.** Pre-training cost measured as theoretical GFLOPs per step and total GPU hours under our setup. BYON incurs higher cost than DiffMAE/MaskDiT due to the added contrastive/uncertainty pathways.

Table 5 reports pre-training cost across methods. BYON requires 63.8 GFLOPs and 97 GPU hours, compared to 45.5 / 84 for DiffMAE and 43.7 / 83 for MaskDiT. Thus, BYON introduces a moderate overhead (40–45% more GFLOPs and 15–17% more wall-clock) relative to diffusion-only baselines, attributable to the contrastive branch and uncertainty-guided weighting. As shown in the accuracy tables, this extra compute coincides with higher mean performance while maintaining low variance, indicating a favorable accuracy–compute trade-off under our standardized protocol.

# H    EFFECT OF IAN ACROSS MASK RATIOS

|  | Mask Ratio = 50% | Mask Ratio = 60% | Mask Ratio = 70% |
|---|---|---|---|
| w/o IAN | 82.25 | 82.84 | 82.77 |
| w/ IAN | **83.04** | **83.56** | **83.54** |

Table 6: **Effect of IAN across mask ratios.** IAN consistently improves recognition accuracy for 50–70% masking.

Table 6 shows that IAN consistently improves performance across masking levels: +0.79 at 50% (82.25 to 83.04), +0.72 at 60% (82.84 to 83.56), and +0.77 at 70% (82.77 to 83.54). The gains are stable (0.72–0.79) and insensitive to the masking hyperparameter, indicating that saliency-aware corruption complements both moderate and aggressive masking. Intuitively, as masked area increases, reconstruction pressure grows; IAN steers noise toward informative content, yielding more semantically useful denoising signals and thereby tighter transfer performance across ratios.

# I    COMPARISON WITH OFF-THE-SHELF UNCERTAINTY

We replace SUE with an off-the-shelf uncertainty module, DUQ (Deterministic Uncertainty Quantification), keeping all other settings fixed. DUQ attains 83.02%, below SUE's 83.56%. DUQ estimates uncertainty in a task-agnostic manner, whereas SUE is task-coupled: it measures reliability with respect to the reconstruction objective that produces the very features used for alignment. This coupling enables reliability-aware reweighting of the contrastive loss precisely when reconstructions are faithful and de-emphasizes uncertain cases. In contrast, the DUQ signal is less aligned with reconstruction fidelity, yielding weaker calibration for the alignment target and smaller gains. Empirically, SUE provides a stronger global-alignment prior with minimal overhead. In short, for representation transfer, uncertainty must be tied to the reconstruction task (SUE), rather than estimated in a task-agnostic fashion (DUQ).

# J    ONE DECODER VS. TWO DECODERS

We compare a single shared decoder (for both de-noising and de-masking) with two separate decoders (one per task). Two decoders yield 83.59%, only a +0.03 improvement over the shared-decoder configuration (83.56%). Splitting decoders increases parameters and compute but offers limited benefit. The tasks are synergistic at the representation level—sharing a decoder encourages feature reuse and mitigates overfitting to task-specific idiosyncrasies. With separate decoders, the

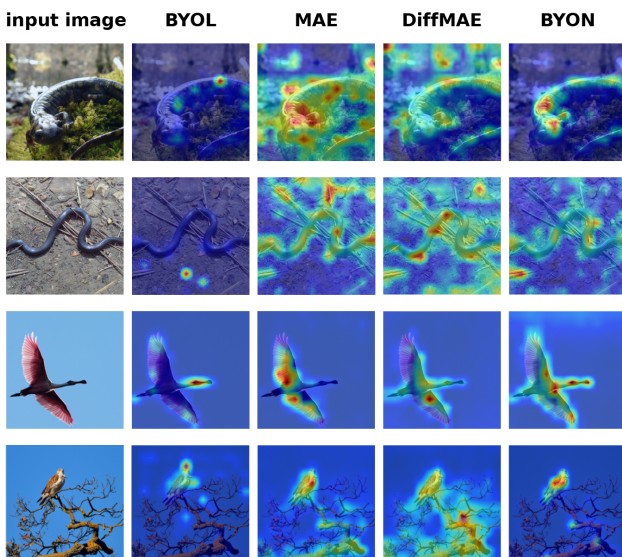

Figure 10: **Qualitative Comparison of Self-Attention Maps.** Visualization of attention maps from models pre-trained with different objectives. MAE (De-masking only) and DiffMAE (De-noising only) shows localized attention focused on reconstruction details, sometimes missing the holistic object context. BYOL (Contrastive only) exhibits broad, global attention but lacks local precision. BYON (Ours) achieves an effective balance, demonstrating both global coherence (holistic object capture) and local discriminative power (focus on salient, fine-grained features).

added capacity does not translate into materially better transfer, likely because the bottleneck is upstream (encoder + alignment) rather than in the decoding head. Moreover, a shared decoder enforces a mild multi-task regularization, stabilizing training without sacrificing accuracy, since both branches ultimately solve the same reconstruction task. Thus, given the marginal accuracy gain and higher complexity, a single decoder is a more favorable design point for recognition-oriented pre-training.

## K   QUALITATIVE ANALYSIS OF PRE-TRAINING OBJECTIVES

To provide deeper qualitative insight into how different pre-training objectives influence the learned representations, we visualize the self-attention maps, following methodologies similar to [1, 2]. We compare three distinct approaches pre-trained under identical settings: BYOL (Grill et al., 2020) (Contrastive only), MAE (He et al., 2022) (De-masking only), DiffMAE (Wei et al., 2023) (De-noising only, where noised patches replace masked patches), and the proposed BYON.

As illustrated in Figure 10, the attention maps reveal distinct patterns corresponding to the inductive biases of each method. MAE (De-masking only) and DiffMAE (De-noising only) display localized and sometimes fragmented attention. The model focuses intensely on specific textures, edges, or high-frequency details that are critical for the pixel-level reconstruction task. While this demonstrates strong local feature learning, the attention often fails to capture the holistic semantic context or the complete boundaries of the object, confirming the local bias observed in Figure 1(b).

Conversely, the attention maps for BYOL (Contrastive-only) tend to exhibit broad attention fields that cover the extent of the main objects. This aligns with the instance-discriminative objective of contrastive learning, which encourages capturing the global structure necessary to distinguish one image from another. However, the attention often lacks precision in highlighting fine-grained, local details, appearing somewhat diffuse over the object's surface.

BYON demonstrates a superior balance between these two extremes. The attention maps are globally coherent, clearly delineating salient objects within the scene. Simultaneously, they remain locally discriminative, focusing precisely on the most informative regions and adhering sharply to

semantic boundaries. This result supports the central claim that BYON effectively unifies local and global representation learning. This balance emerges directly from our coupled objective and the self-reinforcing loop. Reliable local reconstruction (guided by SUE) stabilizes the targets for global contrastive alignment, and in turn, improved global alignment enhances the encoder's ability to focus on semantically meaningful structures during the denoising process.

## L DETAILED FORMULATION OF RECONSTRUCTION LOSSES

This section provides the detailed mathematical formulations for the two reconstruction objectives utilized in the BYON framework: the De-masking loss ($\mathcal{L}_{\text{demask}}$) and the De-noising loss ($\mathcal{L}_{\text{denoise}}$). These losses supervise the local feature learning process by encouraging the recovery of the original input signal from corrupted representations.

### L.1 PRELIMINARIES AND NOTATION

Let $I$ be the original input image, tokenized into a sequence of $N$ patches. We denote the normalized pixel values of these patches as $x \in \mathbb{R}^{N \times D}$. In the context of the diffusion process, we refer to these clean input tokens as $x_0$.

We define a binary mask $M \in \{0, 1\}^N$. For the $i$-th token, $M_i = 0$ indicates that the token is masked (replaced by a learnable embedding $\theta$), and $M_i = 1$ indicates that the token is visible. In BYON, the visible tokens are further corrupted by noise.

The decoder network (composed of $\Phi_{\text{demask}}$ and $\Phi_{\text{denoise}}$, as introduced in Equation 11) aims to reconstruct the original input $x$ from the corrupted input sequence $\bar{x}$ (Equation 9).

### L.2 DE-MASKING LOSS ($\mathcal{L}_{\text{DEMASK}}$)

The De-masking objective aims to predict the original content of the masked patches based on the context provided by the visible (albeit noisy) tokens. We employ the Mean Squared Error (MSE) between the reconstructed patches and the ground-truth normalized pixel values. The loss is computed exclusively over the masked tokens.

Let $\hat{x}^{\text{mask}}$ denote the output tokens reconstructed by the de-masking pathway ($\Phi_{\text{demask}}$). The formulation of the De-masking loss is given by:

$$\mathcal{L}_{\text{demask}} = \frac{1}{N_{\text{mask}}} \sum_{i=1}^{N} (1 - M_i) \cdot \|\hat{x}_i^{\text{mask}} - x_i\|_2^2, \tag{13}$$

where $\hat{x}_i^{\text{mask}}$ is the reconstructed representation for the $i$-th token, $x_i$ is the corresponding ground-truth token, and $N_{\text{mask}}$ is the total count of masked tokens, calculated as $N_{\text{mask}} = \sum_{i=1}^{N}(1 - M_i)$. This normalization ensures the loss magnitude is invariant to the masking ratio. This objective forces the model to learn robust contextual representations for infilling missing information.

### L.3 DE-NOISING LOSS ($\mathcal{L}_{\text{DENOISE}}$)

The De-noising objective is based on the principles of Denoising Diffusion Models (DDMs). It trains the model to reverse a predefined forward diffusion process.

#### L.3.1 THE FORWARD PROCESS IN BYON

The forward process gradually corrupts the original data $x_0$ over a sequence of timesteps $T$. This process is defined by a noise schedule that determines the parameters $\alpha_t$. A key property of the diffusion process allows sampling a noisy version $x_t$ at any arbitrary timestep $t$ directly from $x_0$. In BYON, this process is further modulated by the Image-specific Adaptive Noise (IAN) intensity $\eta$ (Section 3.4). The formulation for the visible tokens ($M_i = 1$) is:

$$x_t(x_0, \epsilon, \eta) = \sqrt{\alpha_t} x_0 + \sqrt{1 - \alpha_t}(\eta \odot \epsilon), \quad \text{where } \epsilon \sim \mathcal{N}(0, \mathbf{I}). \tag{14}$$

Here, $\alpha_t$ controls the signal-to-noise ratio at timestep $t$.

### L.3.2 THE TRAINING OBJECTIVE

The goal is to train the decoder network to reverse this process. We adopt the simplified training objective where the network is parameterized to predict the noise realization $\epsilon$ that was added to $x_0$ to produce $x_t$.

Let $\epsilon_\theta(x_t, t)$ denote the noise predicted by the decoder (parameterized by $\theta$, corresponding to the pathway $\Phi_{\text{denoise}}$) given the noisy input $x_t$ and the timestep $t$. The general training objective minimizes the following expectation:

$$\mathcal{L}_{\text{DDPM}} = \mathbb{E}_{t,x_0,\epsilon,\eta} \left[ \|\epsilon - \epsilon_\theta(x_t, t)\|_2^2 \right]. \tag{15}$$

### L.3.3 IMPLEMENTATION DETAILS

In the BYON framework, $\mathcal{L}_{\text{DDPM}}$ is applied specifically to the tokens that are visible (i.e., where $M_i = 1$). During training, a timestep $t$ and noise intensity $\eta$ are determined adaptively via IAN, and the corresponding Gaussian noise $\epsilon$ is sampled and applied.

The De-noising loss is computed as the MSE between the actual Gaussian noise realization ($\epsilon$) and the noise predicted by the decoder ($\epsilon_\theta$) across the iterative refinement:

$$\mathcal{L}_{\text{denoise}} = \frac{1}{N_{\text{visible}}} \sum_{i=1}^{N} M_i \cdot \|\epsilon_i - \epsilon_{\theta,i}(x_t, t)\|_2^2, \tag{16}$$

where $N_{\text{visible}} = \sum_{i=1}^{N} M_i$ is the total count of visible (noised) tokens. This loss implicitly supervises all iterative steps, encouraging the model to reconstruct fine-grained spatial details over progressive refinement.

## M    ADDITIONAL DOWNSTREAM ABLATIONS

In this section, we provide the additional downstream results to clarify the necessity of the diffusion components in BYON. The ImageNet results in Table 1 show that naïvely combining De-masking and De-noising (82.86%) yields no improvement, indicating that uniform denoising added on top of MIM does not produce meaningful gains. This observation provides *strong motivation for our design*: IAN is introduced to make the denoising signal semantically informative, and SUE stabilizes the alignment objective by accounting for reconstruction reliability.

To further validate this argument, we conducted comprehensive downstream ablations on ADE20K (Zhou et al., 2017a) (semantic segmentation) and COCO (Lin et al., 2014a) (object detection). The results clearly show that both IAN and SUE provide consistent improvements across tasks, and that the full BYON model achieves the highest performance in all settings. These downstream results highlight that the benefits of diffusion-based corruption and uncertainty-guided alignment become substantially more pronounced beyond ImageNet classification.

Table 7: Downstream ablations on ADE20K (mIoU) and COCO detection (AP$^{\text{bbox}}$). BYON (Full) consistently outperforms all baselines across tasks.

| Components | ImageNet | ADE20K (mIoU) | COCO (AP$^{\text{bbox}}$) |
|---|---|---|---|
| De-noising | 80.14 | 36.72 | 39.21 |
| De-noising + IAN | 82.38 | 39.95 | 40.83 |
| De-masking | 82.89 | 42.91 | 46.04 |
| De-masking + SUE | 83.02 | 43.33 | 47.19 |
| De-noising + De-masking | 82.86 | 43.02 | 45.62 |
| **BYON (Full)** | **83.56** | **44.69** | **48.54** |

These results confirm that: (i) uniform diffusion noise alone is insufficient, (ii) IAN meaningfully strengthens the denoising path, (iii) SUE reliably enhances global alignment, and (iv) their combination yields additive improvements that manifest strongly in dense prediction tasks.

## N    MULTIMODAL GENERALIZATION ON CLIP ARCHITECTURE

The core principles of BYON—adaptive noise injection based on input importance (IAN) and uncertainty-guided contrastive alignment (SUE)—are architecture-agnostic. Both IAN, which makes the denoising signal semantically informative, and SUE, which stabilizes the alignment objective by accounting for reconstruction reliability, rely on the properties of the learned representation rather than specific modality processing.

To validate the general applicability of our framework beyond vision-only pre-training, we implemented our method on top of the established CLIP (Radford et al., 2021) architecture. The base CLIP model, which inherently handles the joint representation of image and text, serves as a strong foundation for testing cross-modal enhancement. The results, summarized in Table 8, confirm that our principles effectively generalize to a multimodal framework and significantly enhance the joint representation quality.

Table 8: Multimodal Generalization Results on CLIP Architecture.

| Method | CLIP Base | CLIP + BYON |
|---|---|---|
| Top-1 Accuracy (%) | 83.35 | **85.28** |

This result demonstrates a substantial gain of $+1.93\%$, which confirms that our objectives can effectively improve the alignment and representation quality within the complex multimodal space established by the CLIP architecture.

## O    THE USE OF LLMS

LLMs were used only for minor language improvements. They were not involved in the conception of the research, experiments, analysis, interpretation, or drafting.

