# OpenReview forum: "Bootstrap Your Own Noise: Denoising Adaptive Noise in Diffusion Models for SSL"
_ICLR.cc/2026/Conference — Submitted to ICLR 2026_

### Official Review · Reviewer_yQ7K · 2025-10-26

**Soundness:** 3
**Presentation:** 3
**Contribution:** 2
**Rating:** 4
**Confidence:** 5

**Summary:**

This paper, "Bootstrap Your Own Noise" (BYON), introduces a new self-supervised learning (SSL) framework. The core concept is what the authors call a "self-reinforcing loop" that unifies three major paradigms: contrastive learning (like SimCLR), diffusion-based image reconstruction (denoising) and MIM-based image reconstruction (demasking).
To make this work, they introduce two novel modules: a "Semantic Uncertainty Estimation" (SUE) module that adaptively re-weights the contrastive loss (paying more attention to good reconstructions) and an "Image-specific Adaptive Noise" (IAN) module, which cleverly adds more noise to more salient or "informative" parts of the image. They show that the resulting pre-trained model produces very strong features that transfer well to a whole range of downstream tasks like classification, detection, and segmentation.

**Strengths:**

1.	The authors have managed to creatively combine three powerful ideas (denoising/demasking, contrastive, and uncertainty) into a single, cohesive framework. The IAN module, in particular, which adapts the noise schedule based on image content, strikes me as a very smart idea that moves beyond the typical "one-size-fits-all" noise of standard diffusion models.

2.	The evaluation looks impressive.

**Weaknesses:**

1.	This framework has a lot of moving parts: a diffusion model, a contrastive learning branch, an uncertainty module (SUE), and an adaptive noise module (IAN). This has to be an absolute monster to train and tune. The paper indicates that ‘BYON incurs higher cost than DiffMAE/MaskDiT due to the added contrastive/uncertainty pathways’, which I suspect is a major weakness compared to simpler, more scalable methods.

2.	Second, because there are so many new components, it's hard to tell what's really driving the performance. From Table 1, we can see that De-masking + SUE already achieves acc of 83.02%, the introduction of De-noising even make acc a little bit worse. Even with all components, the performance gain seems to be marginal, compared to simply using De-masking.

3.	All the major components including denoising, demasking, contrastive learning and uncertainty estimation are existing and well-explored techniques. The simple combination of these limits the originality of this work.

**Questions:**

See the weaknesses.

---

> ### Author Response · Authors · 2025-11-20
> **Response to Reviewer yQ7K**
>
> **W1: Complexity and Training Cost.**
> The computational overhead is moderate. As detailed in `Appendix G` (Table 5), BYON requires 97 GPU hours vs. 83 hours for MaskDiT (~17% increase). We argue this overhead is well justified by the substantial performance gains (e.g., +1.8% mIoU on ADE20K, +7.5% APmask on COCO over MaskDiT/DiffMAE, Fig 6). Furthermore, `Appendix B` (Fig 8) shows BYON (400 epochs) achieves performance comparable to MAE trained for 800 epochs, demonstrating its efficiency. BYON offers a favorable trade-off between cost and representation quality.
>
> **W2: Marginal performance gain; What drives the performance?**
> We appreciate the opportunity to clarify the ablation results.
>
> As the reviewer observes, on ImageNet (Table 1), naively combining De-noising to De-masking (82.89%) offers no gain (82.86%), indicating that simply adding uniform denoising to MIM does not yield meaningful gains. This provides *strong motivation for our design*, IAN and SUE. IAN makes the denoising signal semantically meaningful, and SUE enhances the semantic alignment of diffusion-recovered features. These two proposed components drive the majority of the performance improvements.
>
> The notable gain of +0.54% on ImageNet, shown by comparing the reviewer-mentioned 'De-masking + SUE' configuration (83.02%) with the full BYON model (83.56%), is particularly significant ***considering the highly saturated and competitive nature of the ImageNet benchmark***.
>
> Crucially, the benefits of BYON (IAN and SUE) become much clearer in downstream tasks. We conducted the additional ablations on ADE20K (Segmentation) and COCO (Detection) in `Appendix M` of the revised manuscript as below:
>
> | Components               | ImageNet | ADE20K (mIoU) | COCO (AP^bbox) |
> |--------------------------|----------|---------------|---------------------------|
> | De-noising               | 80.14    | 36.72         | 39.21                     |
> | De-noising + IAN         | 82.38    | 39.95         | 40.83                     |
> | De-masking               | 82.89    | 42.91         | 46.04                     |
> | De-masking + SUE         | 83.02    | 43.33         | 47.19                     |
> | De-noising + De-masking  | 82.86    | 43.02         | 45.62                     |
> | **BYON (Full)**          | **83.56**| **44.69**     | **48.54**                 |
>
> **W3: Limited originality; combination of existing techniques.**
> We respectfully disagree that BYON is a simple combination. The novelty lies in *how* these established paradigms are integrated to address specific failures in prior SSL work. Unifying generative (diffusion/MIM) and discriminative (Contrastive Leraning) approaches effectively is non-trivial.
>
> BYON introduces *two novel mechanisms specifically designed for this integration* (detailed in `Appendix F`):
> 1) IAN (Image-specific Adaptive Noise): A novel approach in diffusion-based SSL that moves beyond uniform noise by adjusting corruption based on saliency.
> 2) SUE (Semantic Uncertainty Estimation): Provides a novel way to guide contrastive learning based on the reliability of the diffusion reconstruction, enabling a self-reinforcing loop.
>
> This integrated design leads to empirical improvements over methods that use these techniques in isolation or simpler combinations (e.g., DiffMAE [1], CMAE [2]).
>
> ----
> **References**
>
> [1] Diffusion models as masked autoencoders.
>
> [2] Contrastive masked autoencoders are stronger vision learners.

---

> > ### Author Response · Authors · 2025-11-27
> > **Gentle Reminder for Reviewer yQ7K**
> >
> > Thank you again for your thoughtful feedback on our submission. We would like to draw your attention to our responses to your concerns, along with the updated manuscript, as noted above.
> >
> > We would greatly appreciate it if you could take a moment to let us know of any further feedback or unresolved issues at your earliest convenience. *Your thoughtful comments have been instrumental in improving our work*, and we want to ensure that we address any remaining points before the deadline.
> >
> > We look forward to your continued engagement and appreciate your time and effort.
> >
> > Thank you.

---

### Official Review · Reviewer_9LjC · 2025-10-31

**Soundness:** 2
**Presentation:** 3
**Contribution:** 2
**Rating:** 4
**Confidence:** 1

**Summary:**

The authors proposed self-supervised, uncertainty-guided contrastive learning method with diffusion model to improve feature representations. They suggested 3 main platforms SUE, IAN, and bootstrapping representations from the diffusion model. The suggested model developed a self-reinforcing loop: better contrastive alignment enhances reconstructions, which in turn refines global feature learning. This method outperforms existing masked image modeling and diffusion-based approaches across classification, segmentation, detection, and fine-grained recognition tasks(ImageNet top-1 accuracy by up to 0.6% and segmentation by 1.8%).

**Strengths:**

- The overall organization and style of the manuscript is high quality. It's easy to follow and try to explain the suggested concept clearly.

-The idea to use image-specific adaptive noise (IAN) with self supervised learning is promising in contrastive learning; IAN makes diffusion model be hard to discriminate source encoder's feature and reference feature.

**Weaknesses:**

- Application set is limited on vision benchmarks such as image classification, segmentation, and detection. can it be further utilized for multimodal learning or cross-modal  cases? This method relies on saliency uncertainty and it is specialized for vision tasks so it may limit  the broadness of applications.

- algorithmic advancement is limited; such contrastive learning may be biased to training set and believed to be unstable under limited configurations (i.e. limited datasets, small model.). The idea brings contrastive learning with bootstrapping generation from diffusion models and it looks working. However, the computational cost of retraining the representers (source and reference encoders) appears to outweigh the resulting reduction in error.

- lacks of details on self-reinforcing feedback loop. Self feedback loop is a main workhorse about this paper, but I didn't catch detailed mathematical description for that. For example, the type of used semantic guidance and category of uncertainty measure were not explained.

**Questions:**

-The section of self-reinforcing feedback loop limits the details. I would like to know the learning objective for self-reinforcing feedback loop and the network type to be used.  Does it use eq.(3) for their training objective?

-IAN adapts diffusion noise per image using token saliency scores. But the interaction between IAN’s noise schedule and the diffusion timesteps is not fully specified in terms of \lambda, \delta, or \eta. Is it chose in a manner of greedy search?

---

> ### Author Response · Authors · 2025-11-20
> **Response to Reviewer 9LjC**
>
> **W1: Limited application (vision only).**
> The core principles of BYON---adaptive noise injection based on input importance (IAN) and uncertainty-guided contrastive alignment (SUE)---are architecture-agnostic. To validate applicability on Multimodal Pretraining, we implemented our framework on top of CLIP [1] architecture. The results are summarized as follows and can also be found in `Appendix N` of the revised manuscript.
>
> - CLIP [1]: 83.351 / CLIP + BYON: 85.276
>
> This confirms that our principles also generalize to multimodal framework effectively and enhance joint representation quality.
>
> **W2: Limited algorithmic advancement & Computational cost.**
> - **Algorithmic Advancement**: The novelty lies in the *integration strategy*---specifically, the design of SUE and IAN and the resulting self-reinforcing loop. This integration is non-trivial and addresses specific limitations of prior diffusion-SSL methods (e.g., uniform noise, lack of global alignment).
> - **Stability/Bias**: SUE is designed to mitigate instability by down-weighting uncertain reconstructions. This is supported by the low variance observed in our multi-seed evaluation (`Appendix A`, Table 2: 83.56 ± 0.16).
> - **Cost**: Regarding computational cost (`Appendix G`, Table 5), BYON requires 97 GPU hours compared to 83 hours for MaskDiT (~17% overhead). This is justified by the significant performance improvements (e.g., +1.8% mIoU on ADE20K, +7.5% APmask on COCO over MaskDiT/DiffMAE, Fig 6).
>
> **W3 & Q1: Details on the self-reinforcing feedback loop.**
> The self-reinforcing loop is an *emergent property of the framework*, not a separate module.
>
> 1) Uncertainty Measure (SUE): SUE (Section 3.2) estimates reliability using the L1 distance between the reconstructed and original images (Eq. 1).
> 2) Semantic Guidance: The resulting uncertainty score guides the contrastive learning objective (Eq. 3), ensuring only reliable reconstructions contribute strongly to the global alignment.
> 3) Feedback Loop: The contrastive loss updates the Encoder. A better-aligned encoder produces higher-quality reconstructions in subsequent iterations. This improved reconstruction leads to lower uncertainty, which in turn strengthens the contrastive learning signal.
>
> The learning objective for the entire system is the combined loss in Eq. 12.
>
> **Q2: Interaction between IAN noise schedule and diffusion timesteps.**
> The diffusion timestep *t* is drawn from the standard range; we simply make it deterministically adaptive by computing *t* from saliency statistics as in Eq. 7. Likewise, the noise magnitude \eta is a saliency-based modulation of the standard \(\sqrt{1 - \alpha_t }\,t\) term. No greedy search or tuning is required, and the base diffusion scheduling remains intact.
>
> ----
> **References**
>
> [1] Learning transferable visual models from natural language supervision.

---

> > ### Comment · Reviewer_9LjC · 2025-11-26
> >
> > Thanks to the author for replying answers back. I read it and it partially answers my previous concerns. I'm still lean on border reject because of limited novelty & configurations. This claim is also observed from a reviewer yQ7K (too many moving parts => many components are combined without significant considerations of easy tuning).

---

> ### Author Response · Authors · 2025-11-26
> **Clarification on Novelty and Configuration**
>
> We sincerely thank Reviewer 9LjC for **continued engagement** and the **opportunity to clarify the concerns** regarding 1. novelty and 2. configuration ("too many moving parts," "easy tuning").
>
> # 1. Novelty: Integration Strategy, Not Mere Combination
>
> We respectfully emphasize that BYON's novelty lies in the **principled integration strategy** required to unify generative (MIM/Diffusion) and discriminative (CL) objectives---a non-trivial challenge in SSL.
>
> - *The Failure of Naive Integration (The strongest evidence)*
>
> As demonstrated empirically in Table 1, mere combinations fail. Simply adding De-noising to De-masking yields no gain (82.86%), performing slightly worse than De-masking alone (82.89%). The same holds for combining contrastive loss, too. This failure occurs because (a) uniform diffusion noise is often uninformative for representation learning, and (b) bootstrapping CL from unreliable reconstructions destabilizes global alignment.
>
> - *Novelty as Principled Coupling*
>
> BYON's contribution is the introduction of two novel couplings specifically designed to overcome these failure modes (detailed in Appendix F):
>
> (i) **Saliency-Aware Corruption (DDM+IAN)**: IAN transforms the denoising task into semantically meaningful feature reconstruction.
>
> (ii) **Reliability-Aware Alignment (CL+SUE)**: SUE is essential for stabilizing bootstrapping by preventing unreliable reconstructions from corrupting the global alignment target.
>
> It is these specific couplings that enable the novel self-reinforcing loop, unlocking synergy that *naive combinations cannot*. This synergy becomes even more evident in downstream tasks, as highlighted in our response to Reviewer yQ7K (W2).
>
> # 2. Configuration: Adaptivity Reduces Tuning Complexity
>
> Regarding concerns about configuration ("easy tuning"), we respectfully argue that BYON emphasizes **adaptivity**, not brittle configuration.
>
> - *Adaptivity vs. Manual Tuning*
>
> IAN and SUE are adaptive. They dynamically adjust the noise schedule (IAN) and the loss weighting (SUE) based on the input data itself (saliency and uncertainty). This data-driven approach *reduces* the burden of manual configuration compared to static, hand-tuned schedules.
>
> The configuration is robust, not complex. Key hyperparameters (e.g., λ=0.1, τ=0.5) were fixed across all experiments (Sec 3.2/3.5). This adaptivity also contributes to the stability and consistency of training, as evidenced by the low variance in our multi-seed evaluation (Appendix A: 83.56 ± 0.16), which is superior to simpler baselines (e.g., DiffMAE ± 0.33). This consistent training behavior ultimately translates into a notable +0.54% ImageNet improvement---particularly meaningful given the highly saturated and competitive nature of this benchmark.
>
> -------
> We hope these clarifications emphasize that BYON introduces a necessary and principled design to bridge the local-global representation gap. We respectfully ask the reviewer to consider these crucial distinctions regarding our novelty and the robustness of our configuration in the final assessment.

---

### Official Review · Reviewer_468K · 2025-11-01

**Soundness:** 3
**Presentation:** 2
**Contribution:** 2
**Rating:** 4
**Confidence:** 3

**Summary:**

- The paper is a self-supervised learning method that aims to improve previous MIM + diffusion pretraining approaches.
- The authors’ motivation is that earlier MIM + diffusion methods understand local features well but lack global feature understanding.
- To address this, they introduce a contrastive loss along with additional components called the SUE module and IAN.
- Using the proposed SSL framework, which enhances both local and global feature understanding, they achieve improved top-1 accuracy on ImageNet classification and significant gains on downstream tasks such as semantic segmentation (ADE20K), object detection (COCO), and instance segmentation (COCO).

**Strengths:**

The proposed SSL method shows performance improvements in image classification, segmentation, and detection, which demonstrates its effectiveness. (There is a slight improvement in image classification, but a large improvement in detection and segmentation.)

**Weaknesses:**

- The authors could have provided more detail in both the architecture figures and the loss formulations.

1. Figure detail: For example, Figure 2 could have been illustrated more clearly, with notations and loss details included. As currently presented, it is difficult to interpret intuitively.
2. Loss detail: The definitions of L_demask, L_denoise are missing.

- It would be helpful to include qualitative comparisons, such as attention maps or other visual analyses, across MIM, diffusion, and MIM + diffusion. These visualizations could provide intuitive insights into whether different pretraining methods capture complementary aspects of the representation.

**Questions:**

- If both MIM and diffusion pretraining aim to learn local features, what is the advantage of combining them? Do they learn local features even better when used together compared to using only one pretraining method?
- In the proposed BYON framework, global feature learning comes from applying contrastive learning on the ViT CLS tokens. However, this same strategy can also be applied to pure MIM pretraining, since MIM methods also use transformer architectures with CLS tokens. The authors should provide quantitative results showing that applying contrastive learning on CLS tokens only with MIM pretraining is inferior to using MIM + diffusion to justify their claim.
- Looking at Table 1, which ablates the proposed components, de-masking (MIM) alone already achieves strong accuracy (82.89), better than de-noising (diffusion) at 80.14. Using both de-noising and de-masking yields 82.86, which suggests that the performance gain mainly comes from de-masking rather than de-noising. This raises the question of whether combining MIM and diffusion is truly necessary for an effective pretraining objective.
- Since the ablation in Table 1 seems to focus solely on image classification, it’s unclear whether the same trend holds for downstream tasks like detection or segmentation. Could you include quantitative downstream results for the ablated components to support the claim that combining MIM + diffusion yields broader benefits?

I would be willing to increase my rating if the authors provide further results or clarifications that address these points.

---

> ### Author Response · Authors · 2025-11-20
> **Response to Reviewer 468K**
>
> **W1: Details in architecture figures and loss formulations.**
> We thank the reviewer for the helpful suggestion. `Figure 2` of the revised manuscript has been updated to use consistent notations from the main text.
>
> In addition, the detailed definitions of L_{demask} and L_{denoise} have been added to `Appendix L`, where we provide their formulations, following standard practice in MIM and diffusion-based SSL literature [1–3].
>
> **W2: Qualitative comparisons.**
> In `Appendix K` of the revised manuscript, we include a comparison of BYOL [4] (Contrastive), MAE [1] (De-masking), DiffMAE [2] (De-noising), and BYON attention maps. BYON produces globally coherent yet locally discriminative representations, whereas the comparison methods tend to capture either predominantly local or global patterns. This emerges directly from our coupled objective: reliable local reconstruction stabilizes global alignment, and global alignment enhances semantically meaningful structures during denoising.
>
> **Q1: Advantage of combining MIM and diffusion?**
> They learn complementary aspects of local features. MIM (De-masking) excels at capturing high-level semantic context and object boundaries by filling large missing regions. Diffusion (De-noising) focuses on iterative refinement, which helps capture fine-grained textures and low-level details. Combining them yields a richer supervisory signal---when integrated through a principled design as in BYON, where IAN makes the denoising signal semantically meaningful and SUE enhances the semantic alignment of diffusion-recovered features, the combination becomes effective.
>
> **Q2: Contrastive learning applied to pure MIM vs. BYON.**
> We have benchmarked against Contrastive-MIM methods in `Appendix D` (Table 3) of the original manuscript as below.
>
> | Method | Type | ImageNet Acc |
> |--------|-------------------------------|----------------|
> | iBOT [5]  | MIM + Contrastive             | 82.03         |
> | CMAE [6]   | MIM + Contrastive             | 82.89         |
> | BYON   | MIM + Diffusion + Contrastive | 83.56 (+0.67) |
>
> BYON outperforms iBOT [5] and CMAE [6], demonstrating that integrating Contrastive loss with the combination of MIM and diffusion (enhanced by IAN and SUE) yields superior representations compared to applying it only to MIM.
>
> **Q3 & Q4: Necessity of Diffusion based on Table 1; Downstream ablations.**
> This is a crucial point. As the reviewer observes, on ImageNet (Table 1), naively combining De-masking and De-noising (82.86) offers no gain, indicating that simply adding uniform denoising to MIM does not yield meaningful gains. This provides *strong motivation for our design*, IAN and SUE. IAN makes the denoising signal semantically meaningful, and SUE enhances the semantic alignment of recovered features.
>
> Crucially, the benefits of BYON (IAN and SUE) become much clearer in downstream tasks. We provide the requested ablations on ADE20K (Segmentation) and COCO (Detection) in `Appendix M` of the revised manuscript as below:
> | Components               | ImageNet | ADE20K (mIoU) | COCO (AP^bbox) |
> |--------------------------|----------|---------------|---------------------------|
> | De-noising               | 80.14    | 36.72         | 39.21                     |
> | De-noising + IAN         | 82.38    | 39.95         | 40.83                     |
> | De-masking               | 82.89    | 42.91         | 46.04                     |
> | De-masking + SUE         | 83.02    | 43.33         | 47.19                     |
> | De-noising + De-masking  | 82.86    | 43.02         | 45.62                     |
> | **BYON (Full)**          | **83.56**| **44.69**     | **48.54**                 |
>
> ----
> **References**
>
> [1] Masked autoencoders are scalable vision learners.
>
> [2] Diffusion models as masked autoencoders.
>
> [3] Fast training of diffusion models with masked transformers.
>
> [4] Bootstrap your own latent-a new approach to self-supervised learning.
>
> [5] ibot: Image bert pre-training with online tokenizer.
>
> [6] Contrastive masked autoencoders are stronger vision learners.

---

> > ### Author Response · Authors · 2025-11-27
> > **Gentle Reminder for Reviewer 468K**
> >
> > Thank you again for your thoughtful feedback on our submission. We would like to draw your attention to our responses to your concerns, along with the updated manuscript, as noted above.
> >
> > We would greatly appreciate it if you could take a moment to let us know of any further feedback or unresolved issues at your earliest convenience. *Your thoughtful comments have been instrumental in improving our work*, and we want to ensure that we address any remaining points before the deadline.
> >
> > We look forward to your continued engagement and appreciate your time and effort.
> >
> > Thank you.

---

### Official Review · Reviewer_MG4B · 2025-11-01

**Soundness:** 3
**Presentation:** 3
**Contribution:** 3
**Rating:** 6
**Confidence:** 2

**Summary:**

The paper proposes **Bootstrap Your Own Noise (BYON)**, a self-supervised learning framework that combines diffusion-based denoising with contrastive representation learning.
To address the limitations of uniform noise and instability in previous diffusion SSL methods, the authors introduce two components:

1. **Semantic Uncertainty Estimation (SUE)**, which adaptively weights contrastive objectives based on reconstruction reliability.
2. **Image-specific Adaptive Noise (IAN)**, which adjusts the corruption strength according to token saliency, encouraging more informative learning from complex regions.

By coupling local denoising and global alignment objectives, BYON enables mutually reinforcing feature learning.
Extensive experiments on ImageNet, ADE20K, and COCO demonstrate consistent improvements over prior diffusion- and masking-based pretraining approaches.

**Strengths:**

* **Novel integration:** The paper successfully bridges diffusion-based denoising and contrastive learning within a unified self-supervised framework, which represents an important research direction in representation learning. While it remains an open question whether diffusion models can learn representations comparable to or distinct from those obtained through contrastive learning, this work demonstrates that integrating the two paradigms—reconstruction-based methods (e.g., MAE, diffusion) and contrastive learning—can yield complementary benefits and improved performance. This integration is a meaningful and timely contribution.
* **Balanced local–global learning:** The method couples local reconstruction (denoising) with global alignment (contrastive learning), addressing a key gap between generative and discriminative SSL approaches.
* **Comprehensive experiments:** BYON is thoroughly evaluated across classification, segmentation, and detection benchmarks, showing consistent and notable improvements over strong baselines such as MAE, DiffMAE, and MaskDiT.
* **Clarity and reproducibility:** The paper is well-organized, provides detailed ablation studies, and reimplements baselines under consistent training settings, supporting the credibility of the reported gains.

**Weaknesses:**

* **Analysis on self-supervision components:**
  If I understand correctly, the core contribution of this paper lies in integrating *diffusion- and reconstruction-based learning (MAE, denoising)* with *contrastive learning* to jointly learn global and local representations (as shown in Eq. 12).
  However, the ablation study mainly focuses on *de-noising* and *de-masking*, without isolating the effect of *contrastive learning*.
  It would be valuable to analyze how much the contrastive objective itself contributes to the final representation quality.

* **Positioning vs. prior diffusion SSL:**
  The paper could better clarify how BYON conceptually differs from prior diffusion-based SSL methods (e.g., DiffMAE, MaskDiT) beyond empirical performance gains, particularly regarding the training objectives and the resulting representation properties.

**Questions:**

* **Component-wise analysis:**
  Could the authors provide quantitative and/or qualitative comparisons between (1) contrastive-only, and (2) denoising / demasking-only settings?
  Such analysis would highlight how each supervision signal contributes to the learned representation, offering deeper insight into the proposed integration of reconstruction and contrastive paradigms.

* **Qualitative feature visualization:**
  While downstream metrics (e.g., classification, segmentation) demonstrate overall representation quality, it would also be interesting to visualize the feature space directly.
  The self-attention maps in Figure 5 are promising, showing meaningful inter-object similarity.
  Could the authors further visualize feature embeddings, for instance, via PCA or similarity maps as done in DINO[1,2], to illustrate the qualitative differences among features learned with (a) denoising/demasking only, (b) contrastive loss only, and (c) the combined BYON setup?
  Such analysis would strengthen the claim that BYON effectively unifies local and global representation learning.


[1] Emerging Properties in Self-Supervised Vision Transformers

[2] DINOv2: Learning Robust Visual Features without Supervision

---

> ### Author Response · Authors · 2025-11-20
> **Response to Reviewer MG4B**
>
> **W1/Q1/Q2: Analysis of the contribution of the contrastive learning (CL) component.**
>
> **(W1/Q1)** In BYON, the CL component is inherently linked with the Semantic Uncertainty Estimation (SUE) module, as SUE dynamically weights the CL loss (Eq. 3). To fully address the reviewer’s concern, we have additionally included the *Contrastive loss* variant in `Table 1` of the revised manuscript as below.
>
> **Quantitative comparison**:
> - Contrastive loss only (*newly included in the revised manuscript*): 80.27
> - De-noising only: 80.14
> - De-masking only: 82.89
> - BYON: 83.56
>
> As shown in Table 1, the addition of our SUE-guided contrastive objective yields notable gains in every settings, indicating that the contrastive component provides a crucial global alignment signal that complements the local features learned through reconstruction.
>
> **(Q2) Qualitative comparison**:
> In `Appendix K` of the revised manuscript, we include a comparison of BYOL [1] (Contrastive only), MAE (De-masking only), DiffMAE (De-noising only, noised patches replace masked patches), and BYON attention maps. BYON produces globally coherent yet locally discriminative representations, whereas the comparison methods tend to capture either predominantly local or global patterns. This emerges directly from our coupled objective: reliable local reconstruction stabilizes global alignment, and global alignment enhances semantically meaningful structures during denoising.
>
> **W2: Positioning vs. prior diffusion SSL (DiffMAE, MaskDiT).**
> As detailed in `Appendix F` of the original manuscript, the key distinction lies in the training objective itself: diffusion-based SSL methods (DiffMAE [2], MaskDiT [3]) rely solely on reconstruction-driven denoising, focusing on local features. In contrast, BYON introduces a coupled objective (SUE+CL and IAN+DDM) that explicitly integrates local reconstruction with global alignment. We make clear in `Appendix F` that this coupled formulation is the central distinction between BYON and prior diffusion-based SSL methods.
>
> In addition, to clarify the resulting representation properties, we include a visualization-based comparison in `Appendix K` of the revised manuscript. BYON produces globally coherent yet locally discriminative representations, a property that directly arises from our coupled objective.
>
> ----
> **References**
>
> [1] Bootstrap your own latent - a new approach to self-supervised learning.
>
> [2] Diffusion models as masked autoencoders.
>
> [3] Fast training of diffusion models with masked transformers.

---

> > ### Author Response · Authors · 2025-11-27
> > **Gentle Reminder for Reviewer MG4B**
> >
> > Thank you again for your thoughtful feedback on our submission. We would like to draw your attention to our responses to your concerns, along with the updated manuscript, as noted above.
> >
> > We would greatly appreciate it if you could take a moment to let us know of any further feedback or unresolved issues at your earliest convenience. *Your thoughtful comments have been instrumental in improving our work*, and we want to ensure that we address any remaining points before the deadline.
> >
> > We look forward to your continued engagement and appreciate your time and effort.
> >
> > Thank you.

---

### Author Response · Authors · 2025-11-20
**Paper Revision Summary**

We sincerely thank the Area Chair and all reviewers for their time, insightful feedback, and constructive suggestions. We are encouraged by the recognition of our work’s novelty and timeliness (MG4B), the positive assessment of our core ideas (yQ7K, 9LjC), the comments on clarity of presentation (MG4B, 9LjC), and the acknowledgment of thorough and strong empirical results across benchmarks (MG4B, 468K, yQ7K). We also appreciate the confirmation that independent re-implementation supports the credibility of our reported gains (MG4B).

We would like to emphasize that **we have conducted all requested experiments and clarifications**, and have added these results in the Appendix of the revised manuscript. We note, however, that all of our new experiments are **in-line** with our findings, rooted from our initial detailed analysis in the original submission. We believe the new results strengthen our case, but the main substance of the paper, even revised, is the same as before, centered on the key insights we originally presented.

Key revisions (mainly reflected in the Appendix) include:

- **Table 1, Appendix M**: Comprehensive ablation studies and component analysis; Inclusion of the "Contrastive loss only" variant for complete decomposition of learning signals (Table 1), and extended ablations on downstream tasks (ADE20K and COCO) to clarify the contributions of IAN and SUE (Appendix M).

- **Appendix N**: Demonstration of generality and applicability; Extension to Multimodal Pretraining (implemented on the CLIP architecture) to validate the architecture-agnostic nature of the core principles (Appendix N).

- **Appendix K**: Expanded qualitative comparisons and visualizations; Attention map comparisons against BYOL, MAE, and DiffMAE, illustrating the globally coherent yet locally discriminative representations learned by BYON (Appendix K).

- **Figure 2, Appendix L**: Further clarifications and methodological details; Updated architecture diagram with consistent notation (Figure 2) and detailed formulations for the loss functions (Appendix L).

We believe these additions thoroughly address all reviewer concerns and significantly strengthen our manuscript. We thank the reviewers again for their insightful engagement in improving our work.

---

### Meta-Review · Area_Chair_8Pm9 · 2026-01-09

**Summary:**

This paper proposes a unified self-supervised learning framework that integrates diffusion-based denoising/demasking objectives with contrastive learning, aiming to jointly capture local reconstruction and global semantic alignment. The direction itself is timely and relevant, and the paper demonstrates that combining reconstruction-based and contrastive paradigms can yield complementary benefits in certain settings. The method is carefully implemented, the manuscript is well written, and the experimental evaluation is broad, covering classification, detection, and segmentation with consistent training protocols and strong baselines.

Despite these strengths, AC finds that the paper’s core contribution is not sufficiently isolated or convincingly justified, which limits its overall impact.

The central claim is that jointly combining diffusion-style denoising, masked image modeling, contrastive learning, and uncertainty-aware adaptive noise leads to superior representations. However, the ablation results do not clearly support this claim. In several cases, masked image modeling (demasking) alone already accounts for most of the observed performance gains, while adding diffusion-based denoising yields marginal or even negative improvements. The contribution of the contrastive objective is also not cleanly separated, making it difficult to assess whether the reported gains truly arise from the proposed integration rather than from a dominant existing component. As a result, the claimed synergy among the components remains largely speculative.

From a conceptual standpoint, the paper could better clarify how BYON fundamentally differs from prior diffusion-based self-supervised methods such as DiffMAE or MaskDiT beyond empirical improvements. Many of the individual components—denoising, masked reconstruction, contrastive learning, uncertainty estimation, and adaptive noise schedules—are well explored in the literature. While their combination is non-trivial from an engineering perspective, the work does not yet provide a clear new principle of representation learning that emerges uniquely from this integration.

There are also concerns regarding complexity versus benefit. BYON introduces multiple interacting modules, including a diffusion backbone, contrastive branches, uncertainty estimation, and image-adaptive noise scheduling, resulting in substantially higher training and tuning costs compared to simpler baselines. Given that the performance improvements are sometimes modest and not always attributable to all components, the practical trade-off between added complexity and empirical gain is not fully convincing.

Finally, some aspects of the presentation and analysis could be strengthened. Key loss terms and the self-reinforcing feedback loop are not described with sufficient mathematical clarity, and qualitative analyses of learned representations (e.g., feature visualizations) are limited. Moreover, downstream ablations are largely confined to classification, leaving it unclear whether the same component-wise trends hold for detection and segmentation, where the method reports its strongest gains.

In summary, this paper explores a promising and relevant research direction and demonstrates solid experimental performance. However, the lack of clear evidence isolating the benefits of the proposed integration, limited conceptual differentiation from prior work, and increased complexity relative to the gains weaken the overall contribution. AC therefore recommends Reject, while encouraging the authors to further clarify the role of each component and strengthen the conceptual framing in future revisions.

**Reviewer Scores:**

They might be unchanged.

---

### Decision · Program_Chairs · 2026-01-26

Reject